# Bio-Optical Measurements Indicative of Biogeochemical Transformations of Ocean Waters by Coral Reefs

**Arnold G. Dekker** [1,*], **Lesley A. Clementson** [1], **Magnus Wettle** [2], **Nagur Cherukuru** [1], **Hannelie Botha** [1] and **Kadija Oubelkheir** [3]

1   CSIRO Oceans & Atmosphere, Canberra, ACT 2601, Australia; lesley.clementson@csiro.au (L.A.C.); nagur.cherukuru@csiro.au (N.C.); hannelie.botha@csiro.au (H.B.)
2   EOMAP Australia Pty Ltd., Sunshine Coast, QLD 4558, Australia; wettle@eomap.com
3   School of Earth and Planetary Sciences, Faculty of Science and Engineering, Curtin University, GPO Box U1987, Perth, WA 6845, Australia; kadija.oubelkheir@curtin.edu.au
*   Correspondence: arnold.dekker@csiro.au

**Abstract:** The bio-optical properties of coral reef waters were examined across coral reef ecosystems not influenced by land-derived run-off, in the Great Barrier Reef lagoon (Heron Island) and the Coral Sea (the Coringa-Herald and Lihou Reefs). The aim was to determine whether the absorption properties, the concentration-specific absorption properties, and the phytoplankton and non-algal pigmented particle (NAP) absorption concentrations varied from the ocean waters flushing onto the reef at high tide to those waters on the reef or flushing off the reef at low tide. The optical and biogeochemical properties of on-reef waters systematically differed from the surrounding ocean waters. The chl *a* concentration values varied up to 7-fold and the NAP concentrations up to 29-fold; for the reef samples, the chl *a* values were on average 2 to 3 times lower than for the oceans whilst the NAP values were slightly higher on the reefs. The spectral absorption values of the chl *a*, NAP, and colored dissolved organic matter (CDOM) varied up to 6-fold for reef waters and up to 15-fold for ocean waters. The spectral absorption for chl *a* was up to 3-fold lower on the reef waters, the absorption by the CDOM was up to 2-fold higher and the NAP absorption was 1.6-fold higher on the reef waters. The concentration-specific absorption coefficients for chl *a* and NAP varied up to 9-fold in reef waters and up to 30-fold in ocean waters. In the case of Heron Island and Coringa-Herald cays, this concentration-specific absorption was on average 1.3 to 1.7-fold higher for chl *a* and up to 2-fold lower for NAP on the reefs. The Lihou Reef measurements were more ambiguous between the reef waters and ocean waters due to the complex nature and size of this reef. Based on our results, the assumption that the optical properties of on-reef waters and the adjacent ocean waters are the same was shown to be invalid. Ocean waters flowing on to the reef are higher in phytoplankton, whilst waters on the reef or flowing off the reefs are higher in CDOM and NAP. We found differences in the pico,- nano-, and microplankton distributions as well as in the ratios of photosynthetic to photoprotective pigments. The variability in the bio-optical properties between the reef waters and adjacent ocean waters has implications for the estimations of sunlight absorption along the water column, the UV penetration depth, the temperature distributions, and the nutrient and carbon fluxes in coral reef ecosystems. As Earth observation algorithms require proper parameterization for the water column effects when estimating benthic cover, the actual optical properties need to be used. These results will improve the use of Earth observation to systematically map the differences in the water quality between reefs and the adjacent ocean.

**Keywords:** inherent optical properties; concentration specific inherent optical properties; spectral properties; chlorophyll *a*; NAP; CDOM; algal pigments

## 1. Introduction

Coral reefs are found in a wide range of environments, where they provide food and habitats to a large range of organisms as well as many other ecological goods and services.

Warm-water coral reefs occupy shallow, sunlit, warm, and alkaline waters in order to grow and calcify at the high rates necessary to build and maintain their calcium carbonate structures [1]. Despite their importance, coral reefs are facing significant challenges from human activities including pollution, over-harvesting, physical destruction, and climate change [1]. Coral reef waters and the surrounding ocean waters are increasingly being studied for the purposes of understanding coral reef health and resilience, habitat status, primary productivity, nutrient dynamics, and carbon cycling and budgets [2].

Coral reef ecosystems alter the composition of the optically significant constituents in the water column as they are significantly dependent on the surrounding ocean waters bringing them the required sources of food, mainly in the form of phytoplankton and associated biota such as zooplankton as well as bio-available inorganic nutrients and oxygen [1,2. Coral reefs are composed of many benthic organisms that filter the water for food (coral polyps and their zooxanthellae, sponges, crinoids, etc.) and grow on the substratum (macro-algae, seagrasses, coralline algae, and fleshy algae), which may feed higher organisms such as invertebrate, fish, and turtles, etc. For example, sponges are one of nature's richest sources of novel secondary metabolites and have the capacity to disrupt boundary flow as they pump large volumes of seawater into the water column [3]. This seawater is biogeochemically transformed as it passes through the sponge body as a consequence of sponge feeding, excretion, and the activities of microbial symbionts, with important effects on carbon and nutrient cycling and on the organisms in the water column and on the adjacent reef.

Thus, it is a reasonable hypothesis to assume that the biogeochemical transformation of ocean waters coming on to coral reef ecosystems will impact the bio-optical properties of the reef waters draining off the reef. Those ecosystems are under various physical forces ranging from wind or density-driven circulation/currents as well as tidal effects, which can be quite significant in the Great Barrier Reef lagoon and Coral Sea. This research aims to understand whether measurements of optically active substances in ocean water versus water that has been over a coral reef and in its lagoon (e.g., during a tidal cycle) provides us with an understanding and a possible Earth observable tool to determine how much the reefs process phytoplankton from the ocean and transform these into more CDOM and NAP.

Tools to improve our understanding of coral reefs vary from Earth observations for bathymetric and benthic mapping and water quality estimation to underwater light-climate models required for the estimations of primary productivity and as input for biogeochemical modeling [4,5]. All these tools and methods require accurate estimates of the variability in the water column bio-optical properties. Changes in the bio-optical properties affect the sunlight propagation along the water column, thus affecting the temperature distributions and the UV penetration depth; aspects of relevance for understanding the coral reef dynamics.

Many remote sensing studies on coral reef surface water have used satellite ocean color images or in situ measured adjacent-to-coral reef waters to estimate the optical properties of the surrounding ocean waters to then apply to coral reef waters [6–8]. Examples include exploring statistics within the imagery, developing band ratios and indices in an effort to circumvent the difficulties posed by bathymetry and water color, and incorporating site-specific data into supervised classifications. The problem with such approaches is that they tend to be site-specific, sensor-specific, and/or time-specific [9]. An alternative to these image-based approaches is a physics-based, radiative transfer method. The advantages of physics-based inversion methods include the application of one algorithm to a time series of remote sensing data, the ability to simulate the relevance of various sensors to a particular application, and a reduction in the amount of field and laboratory-based measurements [10]. Another advantage is that the same inversion algorithm can be applied across different sensors with different spectral bands and radiometric resolution without requiring new field campaigns to once more establish the empirical relationships via match-ups. In the case of coral reef remote sensing from satellite sensors, a physics-based inversion method

would allow for the temporal and spatial variations in the optical properties of both the reef water and surrounding water and be able to cope with the spatially and temporally (tides) varying bathymetry [11]. We refer to [12] for a detailed overview of Earth observation image inversion methods over coral reefs. A significant conclusion of their work was that the optical properties over reefs need to be properly characterized to be able to perform spectral inversion methods successfully, as was carried out, for example, for near coastal reefs in India [4].

This study is unique in that it only involves reefs that are not influenced by land-derived run-off or other coastal processes. These effects can be quite significant [3,4] and may obscure the differences between the adjacent waters and the on-reef waters.

## 2. Variability in the Optical Properties of Coral Waters

Here, we present an overview of the variability or apparent, inherent, and concentration specific inherent optical properties of coral reef waters to provide context to this study. Adapted after [13]: The intensity and spectral quality of light available at the benthos is determined by the composition and direction of the downwelling irradiance at the water surface, the absorption and (back-) scattering of photons as they travel downward through the water column, and the interactions within the corals and associated floral and faunal structures. These water column processes are defined by the inherent optical properties (IOPs) of the water, which vary according to the composition, concentration, and size of the suspended particles and dissolved material. The magnitude of the measured absorption (a, $m^{-1}$), scattering (b, $m^{-1}$), and beam attenuation (c, $m^{-1}$) coefficients as well as their spectral shapes and relative ratios can be related to the specific biogeochemical and compositional properties of the optically significant constituents, namely, CDOM, NAP, and phytoplankton.

Various studies have reported a significant variability of apparent optical properties (AOPs), IOPs, and specific inherent optical properties (SIOPs) for coral reefs [4,14], fringing reefs, and [13] for Hawaii, Guam, Palau, and portions of the Great Barrier Reef. However, most publications still assume that the ocean waters surrounding reefs and the reef waters are the same and are homogeneous [8,9,15–17]. In [18], the authors pointed out that although the slope of the absorption coefficient of CDOM ($S_{CDOM}$) is highly variable in natural waters, most remote sensing algorithms keep it as a constant or model it empirically. The study in [19] measured a limited set of IOPs for three study sites in the Great Barrier Reef (reporting chl *a* concentrations between 0.05 and 0.35 mg $m^{-3}$), but for their subsequent modeling work, they reverted to using the lowest measured chl *a* concentrations (from their study site with the clearest waters). The authors in [9] defined their coral water as being of the oceanic type, with a chl *a* concentration of 0.5 mg $m^{-3}$, while [20] defined their coral waters by a slightly more sophisticated four component model, where the CDOM co-varied with chl *a*, similar to an ocean-type relationship. However, they did assume independent variability of the carbonate sediment. Their clear reef water was defined as 0.3 mg $m^{-3}$ chl *a* and 0.3 mg $L^{-1}$ carbonate sediment, and the turbid reef water was defined as 1.0 mg $m^{-3}$ chl *a* and 3.0 mg $L^{-1}$ carbonate sediment. In situ IOP measurements by [21] over coral and sand substrates in the Bahamas demonstrated that the optical properties of the water column, particularly within 1 m of the substratum, were related to the bottom composition through biogeochemical processes. They reported on the concentration of chl *a*, the absorption coefficient of CDOM at 440 nm ($a_{CDOM}(440)$), the attenuation coefficient of the total suspended particulate mass at 650 nm as well as the exponential slope of the absorption coefficient of CDOM ($S_{CDOM}$). Overall, they found that the variability in all of the properties was larger over the reef: the CDOM content was larger over the reef; attenuation was higher over sand, but its spectral slope was larger over the reef; and that the chlorophyll *a* concentration was in most cases lower over the reef.

In [16], the authors demonstrated through an Earth observation image inversion experiment over coral reefs that the pixel-to-pixel variability in the water column optical properties was derived to be sufficiently significant when compared to assuming homo-

geneity over a 5∗5 25-pixel area. In [22], a modeling study was undertaken for Earth observation-based coral reef assessments. In their model, the non-pure water fraction of the absorption coefficient at 440 nm ranged from 0 to 0.3 m$^{-1}$, and the particulate backscattering coefficient, $b_{bp}(440)$, ranged from 0 to 0.006 m$^{-1}$. This backscattering range is reasonable: in [23], a $b_{bp}$ of 0.0028 m$^{-1}$ was proposed for the Lee Stocking Island site, Bahamas. They also mentioned that reef environments could be optically dynamic and that empirical data were lacking on the backscattering from calcium carbonate sediment resuspension events. The study in [24] parameterized their Earth observation inversion model for application to a coastal reef and seagrass system where they presumed stable SIOPs but varied the water constituent concentration and IOP ranges. For backscattering, they chose the representation of the magnitude of backscattering at 550 nm. The value of the spectral shape parameter, Y, was taken to be 1.0, representative of coastal waters. They also presumed fixed specific absorption coefficients for phytoplankton normalized at 440 nm. They assumed both $a_{CDOM}(440)$ and $a_{NAP}(440)$ to have the same spectral slopes S of 0.0015 whilst varying their input parameterization concentration ranges as follows: $a_{PHY}(440)$ 0.01, 0.03, 0.05 (m$^{-1}$); $a_{(CDOM + NAP)}440$ 0.01, 0.06, 0.11 (m$^{-1}$); $b_{bp}(550)$ 0.001, 0.005, 0.010 (m$^{-1}$). Although McKinna et al. (2015) parameterized their shallow water inversion model (SWIM) with fixed spectral IOP shapes, they concluded that it was not robust for optically complex mid-shelf and nearshore GBR waters. They proposed to overcome this limitation by dynamically varying the spectral IOP model shapes on a pixel-by-pixel basis. In [17], they assumed a case 1 waters chl *a*/CDOM covariance and limited the chl *a* concentration to 1.0 mg m$^{-3}$ when implementing a modeling approach to tune the water column properties for depth retrievals over various coral reefs. Although the method was successful in some cases, they pointed out that their less accurate retrievals were probably due to a violation of this assumption.

The authors in [25] assumed a coastal (case 2) IOP parameter set when implementing a semi-analytical inversion model on the hyperspectral coral reef data. Both the depth and substratum type were successfully retrieved. In a modeling study, [26] characterized their water column as either "forereef" or "lagoon". Whilst both IOP sets were assigned the same chl *a* concentration (0.12 mg m$^{-3}$), the lagoon set had more scattering, as calcareous sand and CDOM ($a_{CDOM}(440)$: 0.008 m$^{-1}$ and 0.04 m$^{-1}$; calcareous sand: 0.01 and 0.4 mg m$^{-3}$, for the forereef and lagoon, respectively).

The study by [13] undertook extensive measurements in 2016–2017 across Hawaii–Kaneohe Bay, Guam, Palau, and two Great Barrier Reef islands (Lizard Island, with significant land mass, and Heron Island, with a small cay). However, unlike our study, which relied mainly on the direct discrete measurements of the optical properties of the water column (namely phytoplankton, CDOM and NAP absorption coefficients, and chl *a* concentration), they mainly used in situ flow through instruments and therefore had to infer many variables using these measurements (resulting in 172 measurement stations (of a total of 246) used in their analyses). Thee total particulate matter absorption and CDOM absorption coefficients were measured in situ using a WETLabs AC-S and the phytoplankton absorption coefficient was derived by partitioning the particulate matter absorption into a phytoplankton and non-algal pigmented particle absorption. To arrive at the TSM (total suspended matter), they set the $b_{bp}$* at a mean value of $0.0156 \pm 0.009$ m$^2$ g$^{-1}$, with a reference wavelength of 532 nm. To estimate the TSM, they then divided the in situ measured $b_{bp}$ by this estimated value of $b_{bp}$*. An estimate of the in situ chl *a* concentration (mg m$^{-3}$) was made by utilizing a power model chl $a = (a_{(PHY, \lambda)}/0.0132)^{1.0967}$, where $a_{(PHY, \lambda)}$ is the line-height-corrected phytoplankton absorption at 676 nm. Interestingly, they were able to estimate the refractive index of bulk particulates relative to seawater, providing them with information about the composition of the particulates. Rather than conducting in situ measurements offshore off the reef, they derived the NASA Moderate Resolution Imaging Spectroradiometer (MODIS_ average monthly data (4 km pixels) of absorption due to CDOM, detritus and phytoplankton absorption.

As the only reef in [13] that did not have fringing and terrestrial influenced reefs was Heron Island, we only present their Heron Island results here. For Heron Island, their median values for $a_{(CDOM + NAP)}$ for the offshore ocean were 0.03 m$^{-1}$ versus 0.04 m$^{-1}$ for the fore reef, back, and lagoon; for $a_{PHY}$, these values were 0.0172 m$^{-1}$ versus 0.022 m$^{-1}$ and for $b_{bp532}$, they were 0.0028 m$^{-1}$ and 0.0045 m$^{-1}$, respectively (based on Figure 6 in [13]). These values were different to the ones measured on Heron Island in the 2005 fieldwork reported below in Tables 1 and 2a, which showed an average (N = 5) value of $a_{CDOM}$ (440) of 0.0415 m$^{-1}$ offshore to 0.0853 m$^{-1}$ in the lagoon waters and an average of $a_{(PHY)}$ of 0.01075 m$^{-1}$ offshore to 0.0065 in the lagoon waters (adding up to $a_{(CDOM + NAP)}$ of ~0.051 and ~0.092, respectively). This lower $a_{CDOM}$ and higher $a_{PHY}$ offshore in 2005 may indicate some of the variability within the reef and across time (across the tidal cycle, daily, seasonally to yearly etc.). Based on their median values, there was therefore an increase for all three parameters from the ocean to the reef waters. It is relevant to point out that this was conducted under all tidal conditions, with no recording or analyses of the prevailing tide, wind, and water flow directions. Thus, although [13] provides a large and valuable dataset of the optical properties of coral reef waters, it cannot really be used to infer, on a tidal cycle basis combined with possible wind driven current data, how coral reefs "process" phytoplankton-containing ocean waters.

**Table 1.** The statistics for the chlorophyll a and non-algal particulate matter concentrations for the reef and adjacent ocean waters of Heron Island, the Coringa-Herald cays, and the Lihou Reef.

| | Reef | Ocean | Ratio |
|---|---|---|---|
| | **Mean $\pm$ STD (Min–Max)** | **Mean $\pm$ STD (Min–Max)** | **Mean R:O** |
| **Chlorophyll *a* (mg m$^{-3}$)** | | | |
| **Heron Island** | 0.113 $\pm$ 0.090 (0.030–0.217) | 0.386 $\pm$ 0.083 (0.285–0.502) | 0.294 |
| **Coringa-Herald** | 0.039 $\pm$ 0.022 (0.022–0.078) | 0.101 $\pm$ 0.023 (0.087–0.128) | 0.389 |
| **Lihou Reef** | 0.053 $\pm$ 0.015 (0.031–0.074) | 0.106 $\pm$ 0.023 (0.079–0.137) | 0.498 |
| **Non-Algal Particulate (mg L$^{-1}$)** | | | |
| **Heron Island** | 1.232 $\pm$ 0.555 (0.686–1.995) | 1.106 $\pm$ 1.345 (0.177–3.765) | 1.114 |
| **Coringa-Herald** | 0.276 $\pm$ 0.241 (0.020–0.560) | 0.233 $\pm$ 0.303 (0.020–0.580) | 1.183 |
| **Lihou Reef** | 1.470 $\pm$ 0.413 (0.982–2.077) | 1.152 $\pm$ 0.338 (0.657–1.655) | 1.276 |

**Table 2.** (**a**) Heron Island 2004 (**b**) Coringa Herald Cays 2006 (**c,d**) Lihou Reefs 2008 fieldwork and laboratory results for optically active constituent concentrations, and absorption related IOPs and SIOPs. ((**a**): * only one CDOM measurement each available for reef and ocean; (**c**): **including** two intermediate tide samples). $a_{(NAP)}*_{440}$ in units of m$^2$ mg$^{-1}$ and $a_{(PHY)}*_{440}$ in units of m$^2$ μg$^{-1}$. R:O = Reef to Ocean ratio.

| (a) | | | |
|---|---|---|---|
| | Reef | Ocean | Ratio |
| Heron Island | Mean $\pm$ STD (Min–Max) N = 5 | Mean $\pm$ STD (Min–Max) N = 6 | Mean R:O |
| $a_{(NAP)440}$ | 0.0054 $\pm$ 0.0015 (0.0036–0.0072) | 0.0043 $\pm$ 0.0010 (0.0032–0.0055) | 1.274 |
| $S_{(NAP)440}$ | −0.0099 $\pm$ 0.0008 (−0.0108−−0.0087) | −0.0096 $\pm$ 0.0012 (−0.0112−−0.0080) | 1.031 |
| $a_{(NAP)*440}$ | 0.0047 $\pm$ 0.0026 (0.0027–0.0091) | 0.0103 $\pm$ 0.0104 (0.0015–0.0282) | 0.458 |
| $a_{(CDOM)440}$ $S_{(CDOM)440}$ | 0.0231 * −0.0183 | 0.0249 * −0.0168 | 0.925 1.089 |
| $a_{(PHY)440}$ | 0.0107 $\pm$ 0.0065 (0.0042–0.0200) | 0.0313 $\pm$ 0.0046 (0.0275–0.0374) | 0.342 |
| $a_{(PHY)*440}$ | 0.1108 $\pm$ 0.0304 (0.0678–0.1419) | 0.0824 $\pm$ 0.0109 (0.0678–0.0999) | 1.345 |

**Table 2.** *Cont.*

| (b) | | | |
|---|---|---|---|
| | Reef | Ocean | Ratio |
| **Coringa Herald Cays** | Mean ± STD (Min–Max) N = 5 | Mean ± STD (Min–Max) N = 3 | Mean R:O |
| $a_{(NAP)440}$ | 0.0026 ± 0.0008 (0.0018–0.0036) | 0.0016 ± 0.0007 (0.0012–0.0024) | 1.589 |
| $S_{(NAP)440}$ | −0.0117 ± 0.0014 (−0.0130−−0.0094) | −0.0122 ± 0.0021 (−0.0138−−0.0099) | 0.959 |
| $a_{(NAP)*440}$ | 0.0291 ± 0.0358 (0.0042–0.0888) | 0.0291 ± 0.0299 (0.0022–0.0613) | 1.001 |
| $a_{(CDOM)440}$ | 0.0853 ± 0.0215 (0.0510–0.1066) | 0.0415 ± 0.0065 (0.0367–0.0489) | 2.055 |
| $S_{(CDOM)440}$ | −0.0105 ± 0.0017 (−0.0125−−0.0079) | −0.0125 ± 0.0013 (−0.0136−−0.0111) | 0.840 |
| $a_{(PHY)440}$ | 0.0065 ± 0.0030 (0.0033–0.0098) | 0.0107 ± 0.0012 (0.0093–0.0116) | 0.608 |
| $a_{(PHY)*440}$ | 0.1784 ± 0.0802 (0.1218–0.3197 | 0.1081 ± 0.0220 (0.0871–0.1310) | 1.650 |

| (c) | | | |
|---|---|---|---|
| | Reef | Ocean | Ratio |
| **Lihou Reefs** | Mean ± STD (Min–Max) N = 6 | Mean ± STD (Min–Max) N = 7 | Mean R:O |
| $a_{(NAP)440}$ | 0.0247 ± 0.0122 (0.0055–0.0353) | 0.0222 ± 0.0164 (0.0044–0.0522) | 1.113 |
| $S_{(NAP)440}$ | −0.0099 ± 0.0004 (−0.0107−−0.0095) | −0.0098 ± 0.0005 (−0.0105−−0.0091) | 1.010 |
| $a_{(NAP)*440}$ | 0.0172 ± 0.0096 (0.0036–0.0321) | 0.0208 ± 0.0144 (0.0032–0.0479) | 0.824 |
| $a_{(CDOM)440}$ | 0.1354 ± 0.0591 (0.0413–0.2190) | 0.1287 ± 0.0892 (0.0092–0.2767) | 1.052 |
| $S_{(CDOM)440}$ | −0.0135 ± 0.0016 (−0.0152−−0.0109) | −0.0152 ± 0.0043 (−0.0243−−0.0123) | 0.888 |
| $a_{(PHY)440}$ | 0.0697 ± 0.0203 (0.0391–0.0880) | 0.0706 ± 0.0327(0.0318–0.1245) | 0.988 |
| $a_{(PHY)*440}$ | 0.0282 ± 0.0088 (0.0154–0.0422) | 0.0401 ± 0.0173 (0.0138–0.0686) | 0.703 |

| (d) | | | |
|---|---|---|---|
| | Reef | Ocean | Ratio |
| **Lihou Reefs** | Mean ± STD (Min–Max) N = 5 | Mean ± STD (Min–Max) N = 6 | Mean R:O |
| $a_{(NAP)440}$ | 0.0233 ± 0.0131 (0.0055–0.0353) | 0.0172 ± 0.0105 (0.0044–0.0300) | 1.358 |
| $S_{(NAP)440}$ | −0.0099 ± 0.0005 (−0.0107−−0.0095) | −0.0098 ± 0.0006 (−0.0105−−0.0091) | 1.010 |
| $a_{(NAP)*440}$ | 0.0142 ± 0.0069 (0.0036–0.0215) | 0.0163 ± 0.0089 (0.0032–0.0263) | 0.868 |
| $a_{(CDOM)440}$ | 0.1187 ± 0.0477 (0.0413–0.1623) | 0.1040 ± 0.0666 (0.0092–0.1554) | 1.141 |
| $S_{(CDOM)440}$ | −0.0141 ± 0.0011 (−0.0152−−0.0127) | −0.0149 ± 0.0047 (−0.0243−−0.0123) | 0.946 |
| $a_{(PHY)440}$ | 0.0698 ± 0.0226 (0.0391–0.0880) | 0.0616 ± 0.0246 (0.0318–0.0864) | 1.134 |
| $a_{(PHY)*440}$ | 0.0254 ± 0.0062 (0.0154–0.0318) | 0.0354 ± 0.0130 (0.0138–0.0508) | 0.718 |

## 3. Study Areas and Measurements

The reef systems reported in this study are Heron Island in the southern Capricorn Bunker Group of the Great Barrier Reef (Figure 1a), the two Coringa-Herald cays in the Coral Sea (Figure 1b), and the Lihou Reef system farther out into the Coral Sea (Figure 1c). Heron Island (23°27′S, 151°55′E) is approximately 80 km east of the Australian mainland, 765 km SSW of the Coringa-Herald cays, and 670 km south of the Lihou Reef. The Coringa-Herald cays (17°00′S, 149°10′E) are 290 km east of the outer Great Barrier Reef into the Coral Sea and 360 km east of the Australian coast. The Lihou Reef (ranging from 18°00′S to 17°10′S and from 151°20′E to 152°10′E) are 280 km east of the Coringa-Herald cays, 550 km

east of the outer Great Barrier Reef into the Coral Sea, and 610 km from the Australian coast. The fieldwork on Heron Island took place from 19–24 May 2004; the Coringa-Herald cays from 29 November to 06 December 2006; and on the Lihou Reef from 5–16 December 2008.

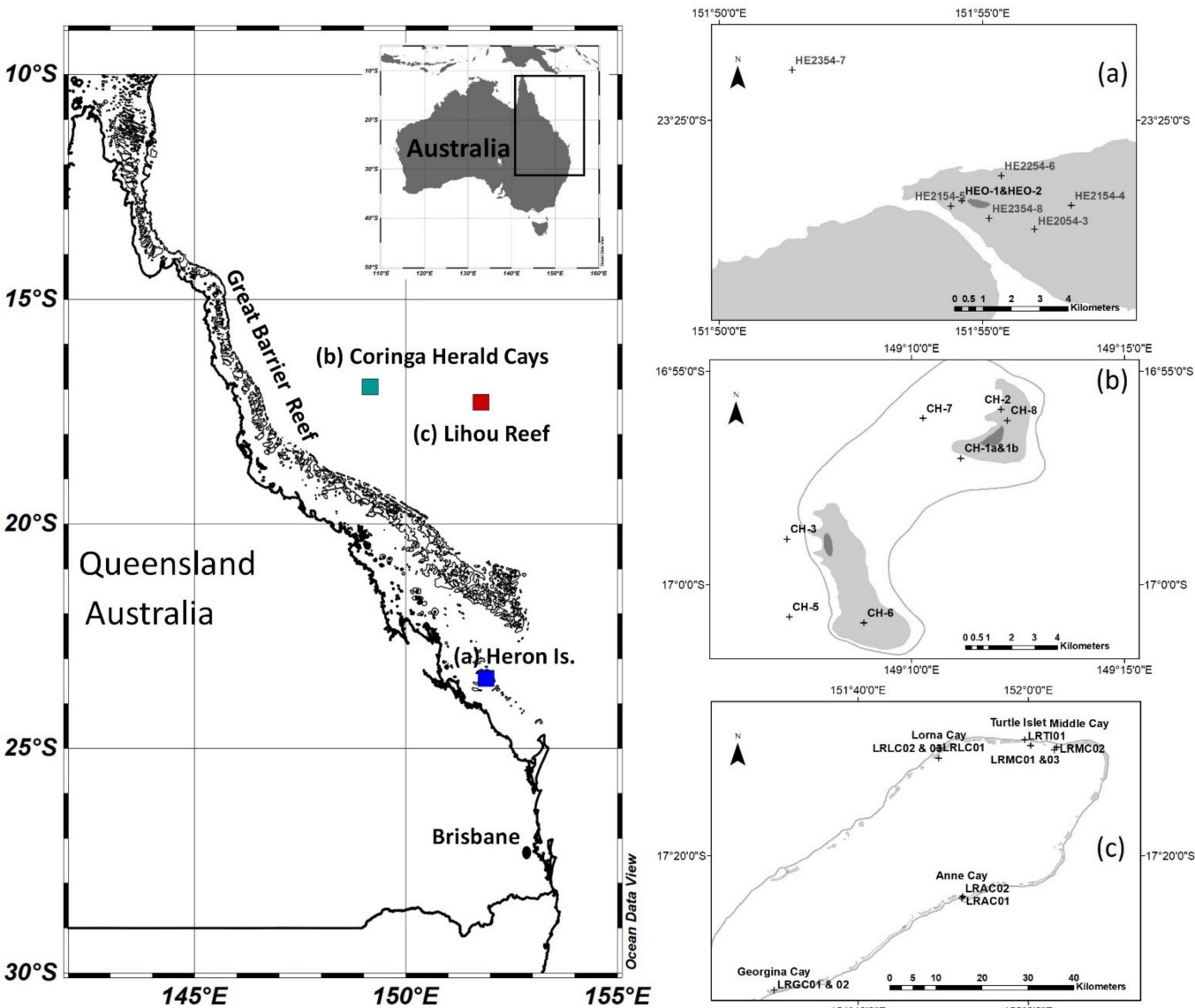

**Figure 1.** An overview of the study area locations of Heron Island 2004, Coringa-Herald Cays 2006, and Lihou Reef 2008 (**left**) and more detailed maps of the reef sample locations (**a**–**c**).

The aim of these fieldtrips was to collect in situ data to develop Earth observation inversion model parameterizations, for example, validation of the substratum and bathymetry mapping, water column composition mapping, and to assess the potential to estimate coral bleaching. The results presented here focus on the assessment of the variability of the water column bio-optical properties above and adjacent to coral reefs. The sampling was carried out from small surface boats near the surface using clean plastic water bottles that were rinsed twice with ambient water before taking a sample. For each fieldwork period, the weather conditions were calm with light winds and for the ocean sampling, only a low swell was present.

### 3.1. Heron Island

The cay itself is mostly surrounded by beach sand. To the east of the cay is the extensive, sandy lagoon, which contains scattered coral bommies and has average depths in the order of 1–5 m. The lagoon has a deeper area to the northeast, and a shallow area to the south. To the west of the island, at the western tip of the reef system, are shallow coral beds. The tidal range is from 1–2 m for the neap and spring tides, respectively. In the open water outside the reef crest, the swell often reaches 3 m. The island is protected from this wave energy by the reef crest, which is mostly composed of live coral. Progressing from the reef crest into the open waters is the reef slope, which commonly features abundant live coral down to 10 m or more. Several sets of both coral reef and open water measurements were collected across eight sites. By sampling at Blue Pools (sample site HE2254_6 Figure 1a) throughout the rising and falling tide, we were able to register both the open ocean waters as well as the reef flat/lagoon waters without having to take the boat inside the inaccessible lagoon. Heron Island can be influenced by diluted terrestrial river output (mainly CDOM) in extreme circumstances due to monsoonal or cyclonic rain. The nearest significant river mouth is ~95 km away (The Fitzroy River). Prior to and during this fieldwork, no such event had taken place, and thus Heron Island may be considered as not being influenced by terrestrial run-off in this case.

### 3.2. Coringa-Herald

Fieldwork was undertaken at the Coringa-Herald Cays (part of the Coringa-Herald Marine Reserve, see Figure 1b) as part of a project to evaluate the capabilities of remote sensing to monitor the aquatic and terrestrial environments of the Australian remote marine parks (reported in [27]). The Coringa-Herald Cays are composed of the NE and the SW Cay connected by a 40 m deep carbonate bridge surrounded by 600 m deep ocean water. The Coringa-Herald SW cay is surrounded by sand. The lagoon is 1 to 12 m deep with several significant channels draining the lagoon to the west. The tidal range is from 0.6 to 1.6 m for the neap and spring tides, respectively. The reef slope contains live coral and sponges to approximately 10 m deep, after which sponges become a more significant part of the biota. The reef crest on the east and south sides is composed mainly of encrusting coralline algae due to the high energy wave regime of the Coral Sea, with waves reaching heights of 8 m or more regularly. The reef flat is mainly bare substratum (e.g., coral sand, coral rubble, and coral rock) and turf algae. Some areas on the outer reef are dominated by a mix of coralline algae, some live coral and green algae (*Halimeda*). The NE Coringa-Herald cay is smaller with a smaller crescent shaped lagoon to the east. The Coringa-Herald NE lagoon is shallow, with a similar reef crest to the SW cay and a similar distribution of cover types. The back slopes of both reefs are dotted with bommies from a few meters to 10 m high. Eight stations were sampled, representing both open water and lagoon environments at both the northeast and southwest Coringa Cays. Water samples in the shallow reef lagoon area were taken from an inflatable boat. During this field study, the horizontal visibility varied from 15 m to more than 40 m between the shallow and deep open waters.

### 3.3. Lihou Reef

Fieldwork was undertaken in the Lihou Reef National Marine Park (Figure 1c) in the Coral Sea (reported in [27]). The Lihou Reef cays are located on the rim of a large carbonate platform of about 110 by 40 km with an average 50 to 65 m deep platform. The tidal range is from 0.6 to 1.6 m for the neap and spring tides, respectively. The platform contains several hundred bommies that range from 45 m deep to the surface (all located within the outer cays). The outer cays are surrounded by 600 m deep Coral Sea water. Measurements were taken of the optical properties of the waters within the Lihou lagoon; the individual cay lagoons of the Georgina, Anne, Lorna, and Turtle cays as well as the surrounding open ocean locations (Figure 1c). For example, Georgina Cay has a reef flat dominated by bare substratum, turf algae and *Halimeda*, and some coral cover; the front reef consists mainly of algae or live coral; while the cay and back reef area are characterized by sand banks,

bare substratum, and some coral and turf algae and *Halimeda*. During this field study, the horizontal visibility varied from approximately 12 to 30 m between the shallow and deep open waters.

### 3.4. Sampling and Laboratory Measurements

For all three fieldwork campaigns, measurements were taken of the HPLC-derived algal pigments, total suspended matter, absorption coefficients of phytoplankton ($a_{PHY}$), non-algal particulate matter ($a_{NAP}$), and colored dissolved organic matter (a(CDOM)). The water samples were filtered following the ESA (European Space Agency) MERIS Cal/Val protocols within 6 h of collection (reference).

### 3.5. Pigment Analysis

The sample water was filtered through a 47-mm glass-fiber filter (Whatman GF/F) using low vacuum and then stored in liquid nitrogen until analysis. The pigment extracts were analyzed using a HPLC method and photo-diode array detection. For the Heron Island samples, the separated pigments were detected at 436 nm and identified against standard spectra using Waters Millenium software. The concentrations of chl *a*, chl *b*, and β,β-carotene were determined from standards (Sigma), and all other pigment concentrations were determined from the standards of purified pigments isolated from algal cultures. A more detailed description of the method can be found in [28]. For the Coringa-Herald and Lihou Reef samples, the separated pigments were detected at 436 nm and identified against standard spectra using Waters Empower software. The concentrations of pigments were determined from commercial and international standards (Sigma, USA; DHI, Denmark). A more detailed description of the method can be found in [28].

### 3.6. Particulate and Non-Algal Absorption

The sample water was filtered through a 25-mm glass-fiber filter (Whatman GF/F) and then stored flat in liquid nitrogen until analysis. The optical density (OD) spectra for the total particulate and non-algal matter were obtained using a GBC 916 (Heron Island and Coringa-Herald samples) and a Cintra 404 (Lihou Reef samples) UV/VIS dual beam spectrophotometer both equipped with an integrating sphere. The pigmented material was extracted from the sample filter [29] to determine the optical density of the non-algal matter The OD spectrum of the phytoplankton pigment was obtained as the difference between the OD of the total particulate and detrital components. The optical density scans were converted to absorption spectra by first normalizing the scans to zero at 830 nm and then correcting for the path length amplification [30]. The total particulate absorbance scan was smoothed using a running box-car filter with a 10-nm width.

### 3.7. CDOM Absorption

In the field, the samples for the CDOM analysis were filtered through a 0.2-µm Durapore filter (Millipore) and stored in acid washed glass bottles and kept cool and dark until analysis. Prior to analysis, the samples were stored under subdued light until they reached room temperature (3–4 h). The CDOM absorbance was measured in a 10-cm path length quartz cell, from 200–900 nm, using the normal cell compartment of the GBC (the Heron Island and Coringa-Herald samples) and a Cintra 404 (Lihou Reef samples) UV/VIS spectrophotometer, with Milli-Q water as the reference. Between the sample scans, the reference cell was removed from the spectrophotometer and placed in a room temperature water bath to reduce the temperature effects in the scans. The CDOM absorption coefficient ($m^{-1}$) was calculated using the equation

$$a_{CDOM} = 2.3(A(\lambda)/l)$$

where $A(\lambda)$ is the absorbance (normalized to zero at 680 nm) and $l$ is the cell path length in meters. An exponential function was fitted to the CDOM spectra over the wavelength range from 350 to 680 nm.

Sodium azide was added to the CDOM samples for the Coringa-Herald and Lihou Reef samples as they needed to be stored for up to two weeks before they could be returned to a lab for analysis. According to [31], sodium azide ($NaN_3$) arrests the degradation of CDOM in samples and reduces the variability in the spectral slope during long-term storage. Sodium azide does, however, slightly increase the absorption coefficient of the natural samples and it is therefore recommended that, if possible, the sample is run fresh. As sodium azide was added to all of the samples from the Coringa-Herald and Lihou Reefs, the differences measured between the reef and ocean waters are still valid.

### 3.8. Total Suspended Matter (TSM) Analysis

A known volume of sample water was filtered through a pre-weighed, muffled (450 °C) glass-fiber filter (47 mm Whatman GF/F). The filter was then rinsed with ~50 mL of distilled water to remove any salt from the filter and stored in the cool and dark until analysis. Filters were dried to constant weight at 65 °C to determine the TSM. Technically, NAP or non-algal pigmented particulate matter is equivalent to TSM minus the equivalent dry weight of the algal pigments extracted using organic solvents. For these field campaigns, NAP was calculated from TSM by subtracting $0.07 * chl\ a$ based on [32]. More recently, [33] found, for three classes of pigments found in microalgae, that the phycobiliproteins may be up to 8% of dry weight, carotenoids usually 0.1–0.2% of dry weight, and chlorophylls 0.5–1.0% of dry weight. These are more precise data and in future, these more detailed values may need to be used.

## 4. Results

Over the three reef ecosystems studied, the chl a concentrations showed a 2- to 7-fold variation on the reef and 2-fold variation in ocean waters. The ratio of the average values of reef chl a to ocean chl a varied from 0.29 to 0.49 (Table 1). The range of chl a concentration over the three reef ecosystems was 0.022 to 0.217 mg m$^{-3}$.

The NAP concentrations showed a 2- to 3-fold variation for reef waters and 2.5- to 10-fold variation in the ocean waters. The ratio of the average values of the reef NAP to ocean NAP varied from 1.18 to 2.14. The range of the NAP concentration was from 0.020 to 3.765 mg L$^{-1}$. The highest value occurred in a deep channel between the Heron Island Reef and Wistari Reef during strong wind conditions, thus the origin of the water was unsure but assumed to be oceanic. This high value, likely due to sediment re-suspension, also led to the $a_{(NAP)}*_{440}$ of 0.0015 m$^2$ mg$^{-1}$ (see Table 2a). If this highest NAP value is removed, the NAP varied from 0.020 to 1.655 mg L$^{-1}$ and the corresponding lowest $a_{(NAP)}*_{440}$ of 0.0034 m$^2$ mg$^{-1}$ for oceanic waters for Heron Island.

The $a_{(PHY)440}$ values showed a 2- to 3-fold variation for reef waters and 1.5- to 4-fold variation in ocean waters. The ratio of the average values of reef $a_{(PHY)440}$ to ocean $a_{(PHY)440}$ varied from 0.36 for Heron Island and the Coringa-Herald cays to 1.0 for the Lihou Reef complex. The $a_{(NAP)440}$ values showed a 1.8- to 6-fold variation for reef waters and 1.3- to 13-fold variation in ocean waters. The ratio of the average values of reef $a_{(NAP)440}$ to ocean $a_{(NAP)440}$ varied from 1.11 to 1.59. $a_{(CDOM)440}$ values showed a 1- to 5.5-fold variation for reef waters and 1- to 15-fold variation in ocean waters. The ratio of the average values of reef $a_{(CDOM)440}$ to ocean $a_{(CDOM)440}$ varied from 1.05 to 2.05.

Figure 2a,b show that at 440 nm, CDOM dominates the absorption coefficient (>40%), followed by phytoplankton with NAP contributing less than 15%. The reef CDOM and NAP contributions were higher than the same measurements for the ocean waters, whilst the $a_{PHY}$ (676) was lower over the reefs than over the oceans. At 676 nm, the phytoplankton contribution was dominant except for Coringa-Herald, where CDOM still contributed between 35% and 90% of the total absorption coefficient, whilst NAP was less than 10% of the absorption coefficient.

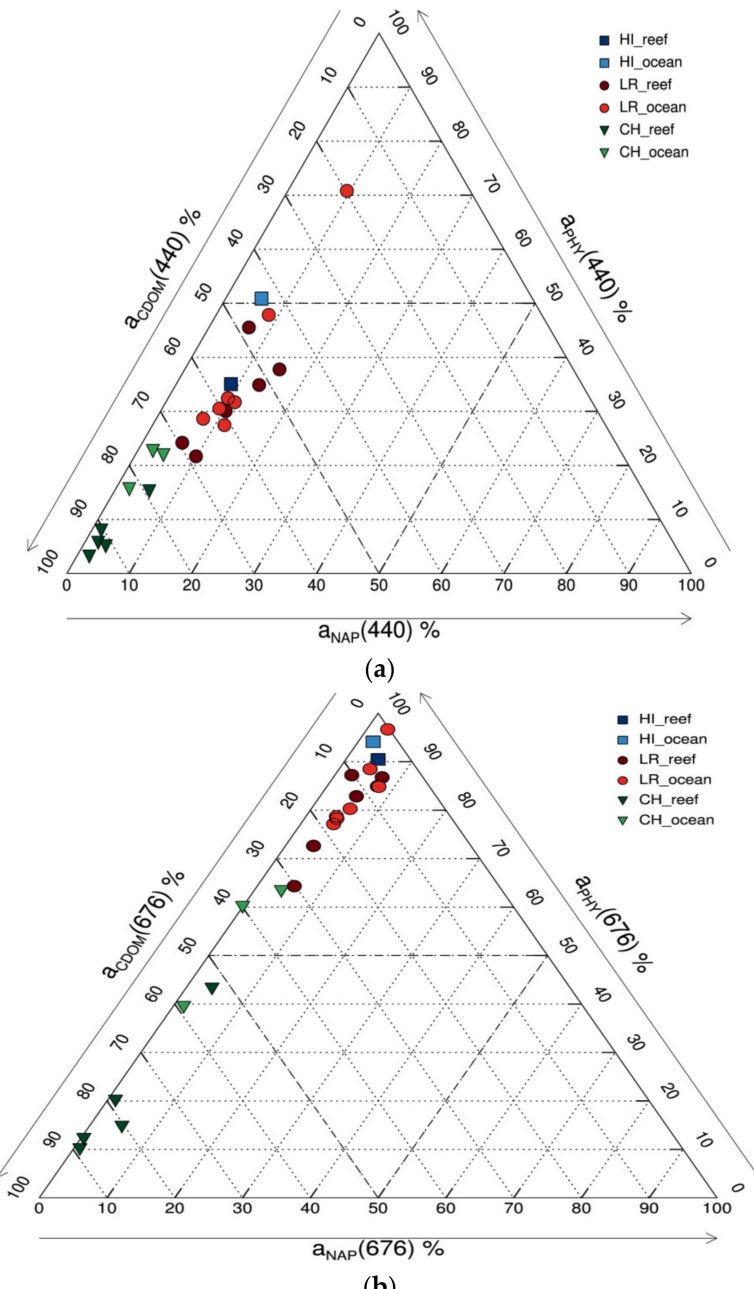

**Figure 2.** (**a**) The absorption budget at 440 nm (the % distribution of absorption at 440 nm by phyto-plankton, NAP and CDOM) for the Heron Island (HI), Lihou Reef (LR), and Coringa-Herald (CH) reef and ocean waters. Note that for Heron Island (HI), there was only one $a_{(CDOM)440}$ measurement each for the reef and ocean waters. (**b**) The absorption budget at 676 nm (the % distribution of absorption at 676 nm by phytoplankton, NAP, and CDOM).

The variability in the concentration specific absorption values $a_{(PHY)}{*}_{440}$ and $a_{(NAP)}{*}_{440}$ was also high. The $a_{(PHY)}{*}_{440}$ values showed a 2- to 3-fold variation for reef waters and 1.5- to 4-fold variation in ocean waters. The reef waters of Heron Island and the Coringa-Herald cays had higher $a_{(PHY)}{*}_{440}$ values, whereas for Lihou Reef, higher values were seen in the ocean waters. The ratio of the average values of reef $a_{(PHY)}{*}_{440}$ to ocean $a_{(PHY)}{*}_{440}$ varied from 1.35 to 1.65 for the samples from Heron Island and Coringa-Herald Cays whilst the Lihou Reef complex had a lower ratio of approximately 0.70. The $a_{(NAP)}{*}_{440}$ values showed a 4- to 9-fold variation for reef waters and a 9- to 30-fold variation in ocean waters. The ratio of the average values of reef $a_{(NAP)}{*}_{440}$ to ocean $a_{(NAP)}{*}_{440}$ varied from 0.46 to 1.00.

The variability in the spectral slopes for CDOM and NAP absorption ($S_{CDOM}$ and $S_{NAP}$, respectively) was also examined between the reef and open ocean waters. Reef waters were characterized on average by lower $S_{CDOM}$ than ocean waters, although for a few samples, higher slopes than those in ocean waters did also occur. $S_{CDOM}$ varied for reef waters from −0.0079 to −0.0183 and for ocean waters from −0.0111 to −0.0243. It is relevant to note that the photobleaching of CDOM may occur over the reefs more than in the ocean waters, which may affect these values. The $S_{NAP}$ values ranged within the range of the $S_{CDOM}$ values. The $S_{NAP}$ varied for reef waters from −0.0087 to −0.0130 and for ocean waters from −0.0080 to −0.0138, indicating that the slope of the NAP absorption may not vary much between the ocean and reef waters.

From these results, we can derive that the optically significant water column constituents varied significantly both on the reef and in the surrounding ocean waters adjacent to the reef. The chl a concentrations in the reef waters were lower on average than for the ocean waters; the NAP and CDOM absorption coefficients at 440 nm were higher in the reef waters. This indicated that, most likely through multiple biogeochemical transformations, the coral reefs and associated biota 'filter' the algae and produce CDOM and NAP. The increase in NAP may be caused by detrital particles from corals and other productive benthic material, by re-suspended sediment, and by coral-eating fish (e.g., parrot fish), etc. However, the similar $S_{(NAP)}$ indicates a similar color of this material, whilst the $a_{(NAP)}*_{440}$ variability between the ocean and reef waters indicates different particle sizes or refractive indices [34]. For the absorption of the three components, an analogous pattern existed, except for Lihou Reef, where the ratio of the mean $a_{(PHY)440}$ values of the reef and ocean waters was equal to 1.

At all three sites, there was greater phytoplankton biomass, as indicated by the chl a concentration, in the ocean waters than in the reef waters (Figure 3a and Table 3). Although the biomass was greater at the ocean water sites, the composition of the phytoplankton community, as determined by pigment analysis, was the same in both water types (Table 3). Cyanobacteria (Synechococcus sp. and Prochlorococcus sp.) dominated at the Coringa-Herald and Lihou Reefs, while a combination of cyanobacteria, haptophytes, and diatoms dominated at the Heron Island sites. The equal dominance of three algal groups at Heron Island could be due to the time of year the sampling took place—winter compared to summer samplings at the other two sites. The size class analysis, based on established methods using diagnostic pigments [35,36], provided the proportions of micro (>20 μm), nano (2–20 μm), and pico-phytoplankton (<2 μm) in the phytoplankton community (Figure 3c and Table 3). At the Coringa-Herald and Lihou Reefs, pico-phytoplankton dominated in the surface layers of both the ocean and reef waters while the nano-phytoplankton decreased and the micro-phytoplankton increased in the reef waters compared to the ocean waters, indicating possible selective phytoplankton group grazing by reef organisms. At Heron Island, pico-phytoplankton dominated in the surface layers of the reef water, but at the ocean water sites, the three size classes were in equal proportions, supporting the equal dominance of cyanobacteria (pico), haptophytes (nano), and diatoms (micro).

Carotenoid accessory pigments are determined to be photosynthetic (PSC), those that transfer energy to reaction centers during photosynthesis or photoprotective (PPC), and those that prevent damage to the chloroplast from high light conditions. Ratios of PPC:PSC greater than 1 indicate a dominance of photo-protective pigments, and such pigments are generally associated with cryptophytes, green algae, and cyanobacteria. A ratio of less than 1 indicates a dominance of photo-synthetic pigments generally associated with diatoms, dinoflagellates, and haptophytes. At the Coringa-Herald and Lihou Reefs (Figure 3d and Table 3), the PPC:PSC ratio was greater than one in both water types, which supports the dominance of the picoplankton cyanobacteria. At Heron Island, the ratio was less than 1 in the ocean waters and slightly greater than 1 in the reef waters. The dominance of PSC in the ocean waters and the small ratio greater than 1 in reef waters is supported by the increased presence of haptophytes (nano) and diatoms (micro) at the Heron Island sites compared to the other two sites.

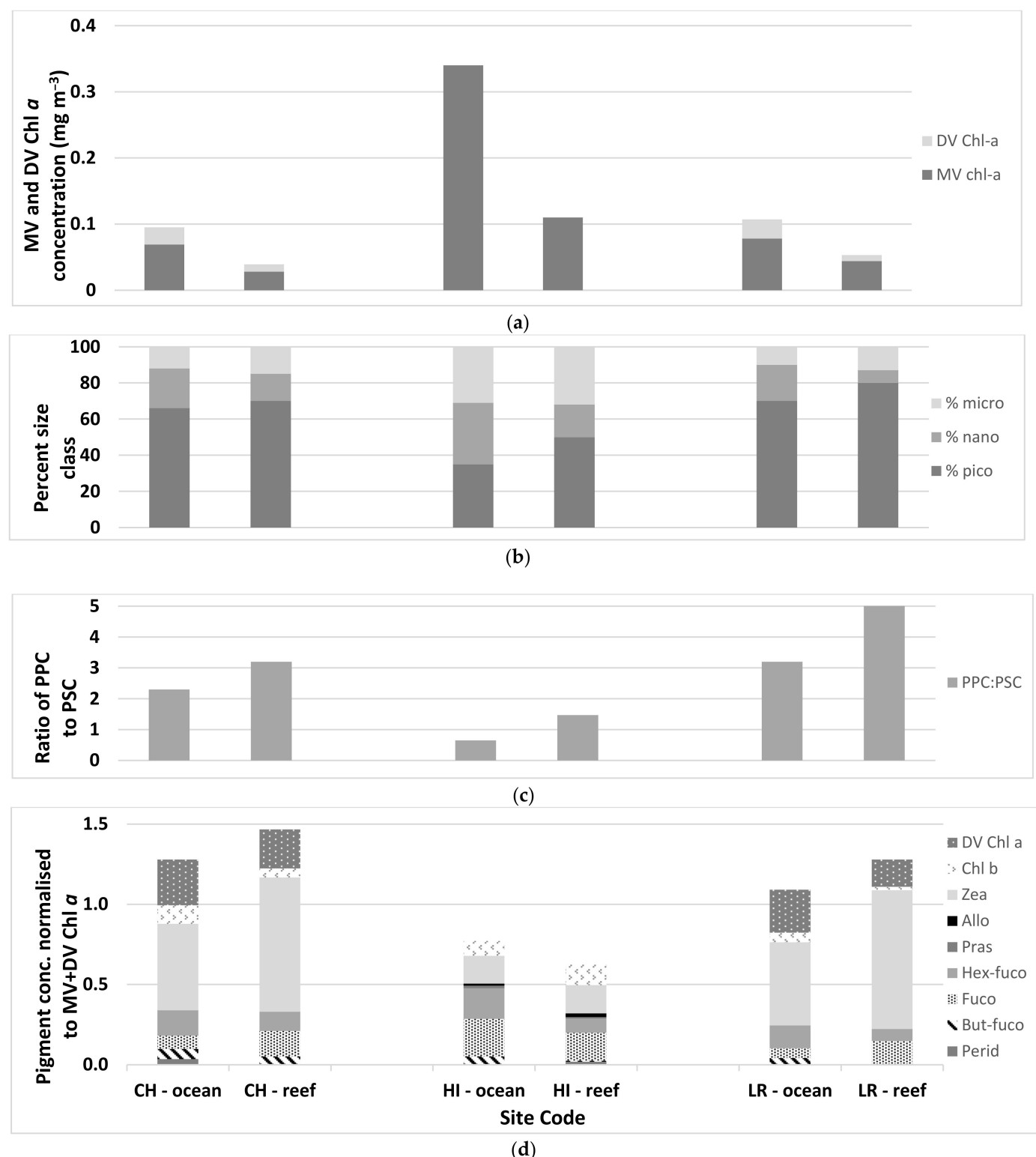

**Figure 3.** A comparison between the reef and ocean sites at Coringa-Herald Reef (CH), Heron Island (HI), and Lihou Reef (LR) for (**a**) average values of concentration of MV and DV Chl *a*, (**b**) percent size class for the pico-, nano-, and microplankton, (**c**) ratio of the photoprotective carotenoids (PPC) to photosynthetic carotenoids (PSC), (**d**) pigment composition based on the pigment concentration normalized to MV+DV Chl *a*.

**Table 3.** The average values for the phytoplankton associated variables measured at the three sites. The dominant phytoplankton group was determined from the pigment composition analysis. PPC—photo-protective carotenoids; PSC—photo-synthetic carotenoids.

| | Total Chl *a* (mg m$^{-3}$) | Dominant Phytoplankton | Picoplankton | Size Class (%) Nanoplankton | Microplankton | PPC:PSC |
|---|---|---|---|---|---|---|
| *Heron Island* | | | | | | |
| Ocean waters | $0.39 \pm 0.08$ | Cyanobacteria, haptophytes, diatoms | $35 \pm 3$ | $34 \pm 4$ | $31 \pm 6$ | $0.7 \pm 0.1$ |
| Reef waters | $0.11 \pm 0.09$ | Cyanobacteria, haptophytes, diatoms | $50 \pm 9$ | $18 \pm 11$ | $32 \pm 4$ | $1.5 \pm 0.6$ |
| *Coringa Herald Reef* | | | | | | |
| Ocean waters | $0.101 \pm 0.023$ | Cyanobacteria | $66 \pm 3$ | $22 \pm 5$ | $12 \pm 4$ | $2.3 \pm 0.6$ |
| Reef waters | $0.039 \pm 0.022$ | Cyanobacteria | $70 \pm 12$ | $15 \pm 8$ | $15 \pm 10$ | $3.2 \pm 1.5$ |
| *Lihou Reef* | | | | | | |
| Ocean waters | $0.106 \pm 0.023$ | Cyanobacteria | $70 \pm 4$ | $23 \pm 4$ | $10 \pm 9$ | $3.2 \pm 0.7$ |
| Reef waters | $0.053 \pm 0.015$ | Cyanobacteria | $80 \pm 4$ | $7 \pm 6$ | $13 \pm 3$ | $5.3 \pm 1.2$ |

The results for Heron Island and the Coringa-Herald cays were more distinct than for the Lihou Reef cays, which might be due to the source of coral waters. It is less clear whether the measurements that were performed on or adjacent to the Lihou Reef cays were on oceanic waters that were modified or unmodified by the Lihou Reef cays and by the entire large and relatively deep (40–60 m) carbonate platform. As an illustration, we may consider a current of 0.5 m s$^{-1}$ going across the Lihou Reef platform: in the approximate 12 h of a tidal cycle, this water will have moved 21.6 km. Considering that the dimensions of Lihou Reef are 110 km length in an ESE to WNW direction and a width of 30 to 40 km in a NNW to SSE direction, it may take a double tidal cycle of 24 h to five tidal cycles for ocean water entering this reef system at one location exiting the reef platform at another location. These variations in current velocity and direction will be more complex, however, it illustrates the fact that it is less easy to identify reef processed waters from ocean water for the Lihou Reef system.

## 5. Conclusions

### 5.1. Variability in Biogeochemical and Optical Properties in Coral Reef vs. Open Ocean Waters

This study provides a set of measurements of optically significant water column constituent concentrations and related absorption properties over a variety of coral reefs and adjacent ocean waters across the Great Barrier Reef and Coral Sea. As the main aim of each of these studies was mapping the bathymetry and the benthos using Earth observation, the bio-optical datasets presented here were intended for parametrization of the forward and inverse bio-optical models for Earth observation purposes. However, we were able to take a sufficient number of measurements, under calm conditions, in reef waters and the adjacent ocean waters to provide a phenomenological study, indicating that the bio-optical properties over reefs are different from the adjacent ocean waters for reefs that are not affected by terrestrial influences. Although this study was limited in scope, an often-made assumption that the absorption related bio-optical properties of coral reef waters can be approximated with open ocean type waters was found to be erroneous. On the optical and biogeochemical levels, waters over the coral beds and inside reef lagoons differed significantly and consistently from the surrounding open waters (see the ratio R:O in Tables 1 and 2).

These results are supported by research in [2], where they showed that strong diel patterns in seawater carbonate chemistry were observed on the Heron Island reef flat: $pCO_2$ ranged from 281 ppm to 669 ppm whereas in the adjacent Wistari Channel from 341–434 ppm, demonstrating the capacity of reef metabolism to dramatically alter the open surface water conditions.

An investigation into the gradients of IOPs between the adjacent open ocean and coral lagoon waters—as a function of tides, wind and circulation driven currents is recommended. To do this properly, it would be good to separate the coral reefs and surrounding waters into those that are not influenced by land run-off or coastal geomorphic features at all (such as in this study) versus those studies that will need to try and separate the island or continental land and run-off effects from the coral reef effects such as the recent study by [4], where they also provided a good literature review of the reef waters affected by land run off.

Overall, the chl *a* concentrations were found to be lower within the lagoon and over the coral beds when compared to the surrounding open waters. The NAP concentrations were found to be overall higher within the lagoon and over the coral beds when compared to the surrounding open waters. The former can be explained by the phytoplankton being grazed by reef organisms, and the latter is probably due to increased re-suspension in the shallower waters of the lagoon. The picoplankton relative fraction increased for the reef to ocean waters, the nanoplankton decreased, and the microplankton was more or less similar. The ratio of photo-protective carotenoids to photo-synthetic carotenoids increased from the ocean waters to the reef waters. These changes could be due to, for example, the selective grazing of phytoplankton by reef organisms.

The chl *a* measurements obtained agree with similar measurements reported in the literature. The concentration values were within the same interval as those found by [21] in the Bahamas and by [19] adjacent to other reefs in the GBR, although maximum values (0.2 mg m$^{-3}$) were slightly lower than those reported in the literature (0.3 and 0.35 mg m$^{-3}$). The published model parameterization assumptions of a clear reef water as defined by 0.3 mg m$^{-3}$ chl *a* and 0.3 mg L$^{-1}$ carbonate sediment, and a turbid reef water as defined by 1.0 mg m$^{-3}$ chl *a* and 3.0 mg L$^{-1}$ carbonate sediment is invalid for chl *a* as this parameter is shown to occur at higher concentrations in adjacent ocean waters than over reefs [20].

The Lihou Reef a(CDOM)$_{440}$ concentrations showed a larger range than those reported by [21] over coral beds. Estimates of the exponential slope of the CDOM absorption encompassed a similar range of values reported by [21] for a site in the Bahamas.

The measurements reported in this study were for one to two week's fieldwork for each reef system only, and it is expected that temporal gradients in the optical properties will occur. Furthermore, not enough samples were acquired for a statistically valid evaluation for each reef system. However, the parameters measured, which were able to be compared to the few values reported in the literature, indicated good agreement with the coral reef waters. The trend of lower chl *a* over corals was separately confirmed by [21], who attributed this to the same grazing mechanism suggested in this study, however, they did not have access to the phytoplankton pigment information of this study.

While acknowledging the small number of samples, the sampling conditions were calm, indicating that the observed trends (e.g., absorption of chl *a*, phytoplankton size classes, and ratio of PPC:PSC), were consistent for coral reef waters versus open waters (i.e., un-ambiguous groupings for the open and coral waters). Finally, the inverse relationship between the NAP and chl *a* concentrations in comparing the coral reef waters to open waters is supported by the biogeochemical mechanisms. It is therefore recommended that ocean type parametrization optical water models should be avoided when simulating the propagation of light in shallow tropical coral reef waters.

### 5.2. Implications for Underwater Light Climate Models for Remote Sensing of Coral Reefs

Earth observation physics-based inversion methods for assessing water column depth, water column composition, and substratum composition (see [11] for a comparison of

methods) require proper parameterization with a representative range of IOPs and SIOPs. It is recommended that radiative transfer modeling of light fields in coral reef waters should use a reef-specific set of inherent optical properties and concentration ranges. For example, if an Earth observation inversion model over a coral reef uses an adjacent ocean parameterization with too much chl *a*, it will influence the Earth observation-based assessment of chl *a* photosynthesizing benthic organisms.

This analysis identified separate optical characteristics for the open waters and coral reef waters. However, it is important to take into consideration the tide, currents, and geomorphological effects on the potential mixing of these two water types when applying physics-based remote sensing to a given study area.

These findings have important implications for the development of the (global and regional) remote physics-based Earth observation of coral reef systems. However, the data were collected over a one to two week period in three specific locations over different seasons. It is suggested that more effort needs to be directed toward the characterization of the optical properties (and their spatial and temporal gradients) of tropical coral reef waters on the reef, in the lagoon, and in the adjacent deeper waters. The characterization of Heron Island, the Coringa-Herald cays, and the Lihou Reef and cay waters presented here represents a first step in this endeavor.

A set of SIOPs need to be defined and the mechanisms for water type mixing understood for the coral reef region(s) studied. Earth observation-based inversion methods applied to daily ocean color images, which explicitly deal with variable IOPs and SIOPs (see [37] for optically deep waters and [11] for optically shallow waters), can image the distribution of these optical properties and thereby provide more systematic observations of our initial results presented here. Indeed, such properly parametrized ocean color data may become a tool for looking at the biogeochemical transformations caused by corals of their surrounding ocean waters.

**Author Contributions:** Conceptualization, A.G.D.; Methodology, M.W., K.O., L.A.C. and H.B.; Software, M.W.; Validation, H.B., N.C. and L.A.C.; Formal Analysis, M.W., L.A.C., K.O., A.G.D. and H.B.; Investigation, M.W., K.O., L.A.C. and H.B.; Resources, A.G.D.; Writing—Original Draft Preparation, A.G.D.; Writing—Review & Editing A.G.D., K.O. and N.C.; Visualization, H.B. and N.C.; Supervision, A.G.D.; Project Administration, A.G.D.; Funding Acquisition, A.G.D. All authors have read and agreed to the published version of the manuscript.

**Funding:** The Coringa-Herald and Lihou Reef projects were funded by, and carried out in collaboration with, the Australian Government Department of Sustainability, Environment, Water, Population, Arts and Community (SEWPAC) and by CSIRO Oceans & Atmosphere. The field campaign was funded by CSIRO Oceans & Atmosphere, Center for Spatial Environmental Research of the University of Queensland, and an ARC Linkage Project 'Integrating Natural Vision and Remote Sensing'.

**Data Availability Statement:** The data in this research is available through CSIRO Oceans and Atmosphere, Canberra, ACT, Australia.

**Acknowledgments:** P. Daniel (CSIRO) prepared all of the instrumentation for the fieldwork. The Heron Island fieldwork was carried by out by M. Wettle and A. G. Dekker in collaboration with S. Phinn, C, Roelfsema, A. Goldizen, and I. Leiper from the University of Queensland. The Coringa-Herald cays fieldwork was carried by out by M. Wettle and A. G. Dekker and the Lihou Reef fieldwork by Y.-J. Park, E.H. Botha, and A. G. Dekker.

**Conflicts of Interest:** The authors declare no conflict of interest.

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
