# Peer review of "Bio-Optical Measurements Indicative of Biogeochemical Transformations of Ocean Waters by Coral Reefs"

_remotesensing, doi:10.3390/rs14122892_

Round 1
Reviewer 1 Report
Dear authors,
Thanks for incorporating all the comments and suggestions made before by this reviewer into this new improved version of the manuscript. The results are presented now in a much clearer way. Also, thanks for clarifying in your comments the reasoning behind why the results were not submitted for publication earlier. Here are some minor edits to consider for the final version:
1. L17 - define the acronym here: ... and non-algal pigmented particle (NAP) absorptions ...
2. L23 same here: ... NAP and colored dissolved organic matter (CDOM) ...
3. L25: you can use NAP here (instead of the whole name) since it was already defined above
4. L30: This statement still seem to be more of a generalization particularly considering the relative small amount of data from this study. Therefore, I suggest modifying it to something like: "Based on our results, the assumption that the optical properties of on-reef waters and the adjacent ocean waters are the same is shown to be invalid."
5. L224: this seems to be a typo: ...used to infer on a tidal cycle by tidal cycle basis ... ?
Thanks
Author Response
publication earlier. Here are some minor edits to consider for the final version:
- L17 - define the acronym here: ... and non-algal pigmented particle (NAP) absorptions ...DONE
- L23 same here: ... NAP and colored dissolved organic matter (CDOM) ... DONE
- L25: you can use NAP here (instead of the whole name) since it was already defined above DONE
- L30: This statement still seem to be more of a generalization particularly considering the relative small amount of data from this study. Therefore, I suggest modifying it to something like: "Based on our results, the assumption that the optical properties of on-reef waters and the adjacent ocean waters are the same is shown to be invalid." DONE
- L224: this seems to be a typo: ...used to infer on a tidal cycle by tidal cycle basis ... ?REMOVED DOUBLE MENTION OF TIDAL CYCLE
Reviewer 2 Report
Overview
This manuscript describes measurements of bio-optical and biogeochemical properties in coral reef regions and more open ocean water near them in the Great Barrier Reef area in Australia. The goal is to determine if the reefs - separated from land effects - had a measurable impact on the biogeochemical properties of ocean water compared to nearby open ocean areas. The authors provide background information on coral reefs and some of the current tools and methods to monitor their health and changes they may undergo. They discuss previous studies that measured optical properties in coral reefs, but point out that most of these studies were conducted near land and the effect of terrestrial inputs could not be ignored. A description of the study areas in and near the Great Barrier Reef follows, along with the sampling and laboratory analyses performed. Most of the analyses were performed on discrete water samples. The results measurements were discussed, with the authors finding chlorophyll concentrations generally about one quarter to one half the concentrations found in the nearby open oceans. They also found non-algal particle and dissolved organic matter concentrations in the reefs were generally higher than the open oceans. The paper concludes with a discussion of the variability of the biogeochemical properties and the implications for monitoring changes in coral reef ecosystems.
Discussion
General Comments
Coral reef ecosystems are found around the world and serve an important role ecological role in the oceans. They are delicate, extremely sensitive to water conditions, and under increasing threat from climate change. Finding methods to monitor changes to coral ecosystems on synoptic scales is hence an important endeavor. This manuscript simply and straightforwardly describes the findings from a set of field measurements to measure the biogeochemical properties in and near coral reefs. The methods are sound and the manuscript is adequately referenced. It is very well written. There are just a couple of minor issues that the authors might want to address.
Specific Points
Technical Content
The details of the measurement were not provided, other than the locations and dates. Are all the water samples collected at the surface? Were there any profile measurements? Temperature and salinity measurements? The results make intuitive sense, but it might be useful to mention the conditions in which the data were collected.
The authors might want to consider including is a short discussion on how the results could be applied to remote sensing methods. How could this data be used in future remote sensing applications for monitoring changes in reef ecosystems?
Presentation
The paper is very well written. The figures are clear and aid the discussion in the text. It is this reviewer’s opinion, however, that references should be named and not simply referred to by the reference number.
Minor Things:
Line 17: ‘..pigmented particle absorptions concentrations…’ should be absorption.
Line 56: This paragraph seems out of place here. Might be better at the end of introduction section?
Recommendation
Overall, I found this to be a clear and straightforward manuscript reporting measurements of biogeochemical properties in coral reefs and the surrounding open ocean areas. It is well-written and relevant. I would encourage the authors to try to connect their results a bit more strongly to remote sensing methods. Otherwise, except for the minor revisions mentioned above, I recommend this for publication.
Author Response
Responses to reviewer 2:
The details of the measurement were not provided, other than the locations and dates. Are all the water samples collected at the surface? YES FROM SMALL BOATS. Were there any profile measurements? NO Temperature and salinity measurements NO? The results make intuitive sense, but it might be useful to mention the conditions in which the data were collected. HAS BEEN ADDED: UNDER CALM CONDITONS FROM SMALL BOATS
The authors might want to consider including is a short discussion on how the results could be applied to remote sensing methods. How could this data be used in future remote sensing applications for monitoring changes in reef ecosystems? HAVE ADDED SENTENCES IN THE FINAL SECTION THAT STRENGTHEN THE RECOMMENDATIONS AND EXPAND THE APLICATION TO USING DAILY OCEAN COLOUR IMAGE IN FUTURE TO FURTHER SYSTEMATICALLY ASSESS THE BIOGEOCHEMICAL TRANSFORMATIONS CAUSED BY CORAL REEFS
Presentation
The paper is very well written. The figures are clear and aid the discussion in the text. It is this reviewer’s opinion, however, that references should be named and not simply referred to by the reference number. WE AGREE WITH THE REVIEWER, HOWEVER THE JOURNAL POLICY IS TO USE NUMBERING.
Minor Things:
Line 17: ‘..pigmented particle absorptions concentrations…’ should be absorption. DONE
Line 56: This paragraph seems out of place here. Might be better at the end of introduction section? MOVED
Recommendation
Overall, I found this to be a clear and straightforward manuscript reporting measurements of biogeochemical properties in coral reefs and the surrounding open ocean areas. It is well-written and relevant. I would encourage the authors to try to connect their results a bit more strongly to remote sensing methods. Otherwise, except for the minor revisions mentioned above, I recommend this for publication.
This manuscript is a resubmission of an earlier submission. The following is a list of the peer review reports and author responses from that submission.
Round 1
Reviewer 1 Report
Dear authors,
Please find my specific comments/edits below:
- L14 – you already mentioned in the previous line that Coringa Herald and Lihou are in the Coral Sea; no need to mention it again in the same sentence.
- L20-21 – please define NAP and CDOM
- L40 – should say: … earth observations of reefs and the adjacent ocean.
- L62-63 – I suggest changing the text to something like: … invertebrates, fishes, sea turtles, and even some marine mammals.
- L65-72 – this whole section on sponges can be reduced substantially as it does not pertain the main topic of the manuscript.
- L74 – what do you mean by “reef-by-reef ecosystems”?
- L77 – should say: … Reef lagoon and the Coral Sea.
- L80 – this line is confusing. What do you mean by: … its lagoon for e.g. a tidal cycle gives us an understanding …?
- L89 – should say: … for understanding corals. (in plural)
- L90-93 – please include some references to sustain this since you are referring to “the bulk of coral reef ocean color remotes sensing studies”
- In the same paragraph: you start with “The bulk of coral reef ocean color remotes sensing studies to date…” but your reference is from 2004??
- L116 & 123-124 – either use caps or not when spelling AOPs, IOPs, etc but be consistent.
- L127 – “Yet most publications still make the assumption that ocean waters…” BUT, only mentions several papers from almost two decades ago (Yamano and Tamura 2004, Su et al 2008, Kutser et al 2003, Hochberg et al 2003, etc.). There are plenty of more up-to-date publications on this same theme that the authors could have used to sustain their statement. I HIGHLY suggest reviewing this paragraph and updating it with more recent references.
- L133 – should say: … concentration of 0.5 …
- L137 – 0.3 mg m-31???
- L153, 154 and through the whole document – the authors indistinctively use different symbology for the parameters measured. For example, they use bbp and bbp for the particulate backscattering coefficient. Same for other parameters: (a(PHY) vs a(phy), S(NAP) vs S(NAP). While this may seem trivial, they should be consistent through the text to avoid creating confusion among their readers. I would also recommend either as part of the manuscript or as a supplement including an abbreviations table.
- L185 – please define MODIS (most people know what it is, but acronyms should be defined the first time they are used in the manuscript)
- L194 – I suggest changing “in the waters from lagoon” to in the lagoon waters.
- L195-197 – The statement “This lower aCDOM offshore and higher aPHY in 2005 may be indicating some of the variability within the reef and across time…” There is a 14 years difference between your data and that of Russel et al (2019), variability is expected!
- L205 – shouldn’t this be Section 3??
- L216-217 – the fieldwork was conducted 14-18 years ago? Why publishing it now?
- Figure 1 – caption: says Lihou Reefs 2086? In the figure, I highly suggest using a satellite image of the study sites instead of a simple map drawing; that way it is much easier to visualize what’s explained in the text. Additionally, the sample sites names within each reef can barely be seen as it stands now. Finally, please add the names of the sites (Heron Island, etc.) to each individual image.
- L293 – previously you defined CDOM as chromophoric dissolved organic matter. Here you are defining it as colored dissolved organic matter. I know it is the same thing, but please be consistent.
- L349 – why this sub-title is in a different color?
- L354 and elsewhere in the text – a minor thing but… there are several places within the text where words are not separated by a space. i.e., matteris instead of matter is.
- L357-361 – I feel like this sentence needs some English restructuring.
- L362 – I assume Results is the 4th section of the manuscript. There is no number associated with it.
- L363-368 – This 1st paragraph looks more like a Table or Figure caption than an actual paragraph. You can use most of this text to the respective figure or table captions and then have a re-worded paragraph here.
- L371 – are the “a” in chlorophyll a bolded? Please be consistent through the text.
- L372 – [Chl a] range was from 0.022-0.21 ug l-1, but where? Reefs or ocean samples? Or both? Please clarify. Also, elsewhere in the text chlorophyll a concentration is defined as mg m-3. Please be consistent.
- L372-373 – This italic sentence seems misplaced, left pending or even a comment from one of the co-authors that was left here.
- Fig 2a &b – This is a good and important figure, but it gives the impression that there were only one sample for HI_reef and one for HI_ocean while the methodology talks about 8 different samples. Where are the others? If this is an average for HI, then why you used an average for this reef and not for the others?
- L423 – please clarify what you mean by “indicating that the colour of the NAP” may not vary much …
- L427 – why is chl a here is italics?
- L439-443 – This paragraph looks more like as being part of the Discussion section than the Results. I suggest moving it accordingly.
- Tables 1-2(a-d) – these are very poorly constructed tables. Table 1 in the 3rd row shows N=?? (Again, as if this was a comment from a co-author left here from an earlier draft version?). The tables would look much more clean if instead of having Reef Reef Reef Reef … just merge these into a single cell. Also, you can do something like:
|
|
Reef Mean ±STD (Min-Max) |
Ocean Mean ±STD (Min-Max) |
Ratio (R:O) |
|
Chl a (mg m-3) |
|
|
|
|
Heron |
0.113 ±0.090 (0.03-0.217) |
And so on … |
|
|
⁞ |
|
|
|
|
NAP (mg l-1) |
|
|
|
|
Heron |
1.232±0.555 (0.686-1.995) |
And so on … |
|
- In tables like 2b-c – please explain why there seems to be no Mean for items like SNAP440 but there is a STD???
- “Table 3.a (Y-axis missing?)” Is this REALLY your Figure caption?? Same for “Tables 3b-d”. Obviously, these are graphs NOT tables. The graphs are in fact missing their respective axes titles. They are also missing the y-axis border and are very poor. Fig 3b legend shows “5 micro” where it should be % micro. Obviously, please define PPC and PSC in the figure caption in Fig 3c.
- L483 – scientific names should be in italics (i.e., Synechococcus). If the intention is to keep these figures, so far, there is no mention of them in the text.
- L503 – “which supports the dominance of picoplankton cyanobacteria.” – How about cryptophytes and green algae? (based on your discussion about the meaning of ratios < or >1 just before this statement). Please clarify.
- Table 4 shows cyanobacteria, haptophytes and diatoms as the dominant phytoplankton but according to the text on lines 495-507, shouldn’t it be dinoflagellates, haptophytes and diatoms (at least for ocean waters at Heron: PPC:PSC <1)?
- L533 – this is one of the main issues of the manuscript. The authors make a generalization about the often-made assumption of corals waters being approximated with ocean type waters when they categorize this as “erroneous”. How can you come up to this conclusion when for some sites and parameters you are only showing ONE data point? Also, data for each site was collected over a single short time period; how about seasonality on these (and pretty much elsewhere in reef waters). What role does it play? The authors have a short discussion on the limitations of their study later on in the manuscript, but this should be clarified further.
- L535 – in Tables 1 & 2 the ratio is shown as R:O, NOT O:R as it is stated here. Please correct this.
- References section – there is not a specific standard format followed here. Please be consistent. Also, it seems other references were added a posteriori and therefore out of place (even with a different font type). Finally, there are references cited in the text which are missing from the list here. i.e., Vidussi et al 2001, Uitz et al 2006
- Finally, where are the statistics to support your results and conclusions?
Reviewer 2 Report
The study is aimed at determining whether the absorption-related inherent optical properties, the concentration-specific inherent optical properties and phytoplankton pigments varied from the ocean waters flushing onto a reef at high tide to those waters on the reef or flushing off the reef at low tide. This idea is of scientific value because emphasis is made between land-based influences and on-reef effects, and the effect of tides is emphasized, which is often ignored in remote-sensing studies of coral reefs. I found the topic of the ms interesting but the ms itself is not well organized, making it difficult for the reader to understand what is the actual purpose of the study: the introduction is too long (including a review study with unclear relevance), results section contains discussion, the acknowledgements section includes author contributions (for which there should be a separate section), and in some parts the text appears to be unfinished. The English style needs much improvement and the introduction lacks references to support several statements. Although the possible application of remote sensing is discussed shortly, I am of the opinion that this ms is hardly suitable for the journal Remote Sensing. Overall the ms gives the impression that it has not been read and corrected by all authors.
Addresses. Too short.
Corresponding author is not mentioned.
Abstract. Too long. Please try to make it more readable.
Keywords. There should be no overlap with title words. Keywords are supposed to be an addition to title words to facilitate the use of search machines.
Line 45. Introduction. I miss information mentioned in the abstract about “This study is unique in that it only involves reefs that are not influenced by land-derived run-off or other coastal processes.” Some examples should be mentioned of studies that include land-based effects, such as Polonia et al. 2015. https://www.sciencedirect.com/science/article/abs/pii/S0048969715304447?via%3Dihub
Line 46. Style. Two sentences starting with “coral reefs” as subject is not recommended. They can easily be combined.
Line 48. Provide references at the end of the sentence.
Line 50. Provide references at the end of the sentence.
Line 56. Coral reefs ecosystems -> coral reef ecosystems
Line 60. Provide references at the end of the sentence.
Line 60. Rephrase “that filter the water to live”. Perhaps “that filter the water for food”?
Line 61. Corals polyps -> coral polyps
Line 61. Many of these organisms are not filter feeders.
Lines 65. This sentence is not relevant. Delete: “Seemingly simple …. novel secondary metabolites”
Line 74. It is unclear what is meant by “the reef-by-reef ecosystems”
Line 84. light climate -> light-climate
Line 90. coral reef ocean color remote sensing studies -> remote sensing studies on coral reef surface water
Line 93. site specific -> site-specific
Line 110-204. I do not understand why there is a need for a literature review study besides the introduction. This is unusual. This is a research article and therefore only text that is relevant for the research should be included in the introduction. The rest can be used in the discussion where that is relevant.
Line 299. Hyphen: a 47 mm glass-fibre filter -> a 47-mm glass-fibre filter
Lines 221-222. Figure caption belongs underneath the figure. The figure should be divided in a,b,c,d and the caption should give more explanation, like overview of the study area (a) and detailed maps of the sample locations (b,c,d)
Figure 1. The text in some maps (right-hand side) is too small.
Line 316. Hyphen: a 25 mm glass-fibre filter -> a 25-mm glass-fibre filter
Line 326. Hyphen: a 10 nm width -> a 10-nm width
Line 329. Hyphen: a 0.2 mm Durapore filter -> a 0.2-mm Durapore filter
Line 332. Hyphens: a 10 cm path length quartz cell -> a 10-cm path-length quartz cell
Lines 363-368. This section should present results and refer to the tables instead of describing what is mentioned by the tables.
Lines 372-373. Unclear sentence; it seems to be reminder to the authors themselves? “Should compare with other reef systems for all the different properties discussed (in discussion).”
Line 378. The caption should describe a little bit more about where the measurements were taken.
Line 381. The caption should describe a little bit more about where the measurements were taken.
Lines 398-405. This section should present results and refer to the figures instead of describing what is shown by the figures.
Line 432. coral eating fish -> coral-eating fish or corallivorous fish
Line 439. Sounds like discussion: “These results are supported by research by Albright et al (2015) ”
Lines 464, 466, 474. Tables have no headers.
Lines 537-538. Here I would expect references of examples of land-based influences
Lines 603-611. Acknowledgements section includes author contributions.
